# Ancient multiplicity in cyclic nucleotide-gated (CNG) cation channel repertoire was reduced in the ancestor of Olfactores before re-expansion by whole genome duplications in vertebrates

**David Lagman** [ID]*, **Helen J. Haines, Xesús M. Abalo** [ID]¤, **Dan Larhammar**

Science for Life Laboratory, Department of Medical Cell Biology, Biomedical Centre, Uppsala University, Uppsala, Sweden

¤ Current address: Science for Life Laboratory, Department of Gene Technology, KTH Royal Institute of Technology, Uppsala, Sweden

* david.lagman@neuro.uu.se

**Data Availability Statement:** The multiple sequence alignments and tree files in Newick format has been deposited in FigShare and is

## Abstract

Cyclic nucleotide-gated (CNG) cation channels are important heterotetrameric proteins in the retina, with different subunit composition in cone and rod photoreceptor cells: three CNGA3 and one CNGB3 in cones and three CNGA1 and one CNGB1 in rods. CNGA and CNGB subunits form separate subfamilies. We have analyzed the evolution of the CNG gene family in metazoans, with special focus on vertebrates by using sequence-based phylogeny and conservation of chromosomal synteny to deduce paralogons resulting from the early vertebrate whole genome duplications (WGDs). Our analyses show, unexpectedly, that the CNGA subfamily had four sister subfamilies in the ancestor of bilaterians and cnidarians that we named *CNGC, CNGD, CNGE* and *CNGF*. Of these, *CNGC, CNGE* and *CNGF* were lost in the ancestor of Olfactores while *CNGD* was lost in the vertebrate ancestor. The remaining CNGA and CNGB genes were expanded by a local duplication of CNGA and the subsequent chromosome duplications in the basal vertebrate WGD events. Upon some losses, this resulted in the gnathostome ancestor having three members in the visual CNGA subfamily (*CNGA1-3*), a single CNGA4 gene, and two members in the CNGB subfamily (*CNGB1* and *CNGB3*). The nature of chromosomal rearrangements in the vertebrate *CNGA* paralogon was resolved by including the genomes of a non-teleost actinopterygian and an elasmobranch. After the teleost-specific WGD, additional duplicates were generated and retained for *CNGA1, CNGA2, CNGA3* and *CNGB1*. Furthermore, teleosts retain a local duplicate of *CNGB3*. The retention of duplicated CNG genes is explained by their subfunctionalisation and photoreceptor-specific expression. In conclusion, this study provides evidence for four previously unknown CNG subfamilies in metazoans and further evidence that the early vertebrate WGD events were instrumental in the evolution of the vertebrate visual and central nervous systems.

available through the DOI: 10.6084/m9.figshare.c. 6170824.v1.

**Funding:** This work was supported by two grants to DLar from Vetenskapsrådet (C0452101) and Carl Tryggers Stiftelse för Vetenskaplig Forskning (CTS 09:210) (https://www.vr.se and https://www. carltryggersstiftelse.se respectively) and one grant to XMA from Stiftelsen Olle Engkvist Byggmästare (468163019) (https://engkviststiftelserna.se). The funders had no role in study design, data collection and analysis, decision to publish, or preparation of the manuscript.

**Competing interests:** The authors have declared that no competing interests exist.

## Introduction

The evolution of the vertebrate eye has captivated the minds of biologists since the early days of evolutionary research [1]. Modern advances in genomics and transcriptomics have helped better understand the molecular underpinnings of vision evolution. Vertebrates have two major categories of photoreceptor cells in their retinae: the cones and the rods. Both cell types use related but distinct signal pathway proteins encoded by genes of several gene families. The discovery that the common ancestor of vertebrates had undergone two rounds (1R and 2R) of whole genome duplication (WGD) somewhere around the split between gnathostomes and cyclostomes led us to hypothesize, and later demonstrate, that many of the gene families encoding the signaling components of the vertebrate phototransduction cascade expanded during these events [2,3], namely the visual opsins [4], transducins [5,6] and phosphodiesterase 6 (PDE6) [7,8]. Additionally, we have shown that the third WGD (3R), in the teleost fish ancestor [9–12], led to further specializations in the teleost eyes [6–8]. Phylogenetic analyses of other components of all steps of the activation of the vertebrate phototransduction cascade have provided further support for 2R as a major contributor of the molecular machinery used for vision in cones and rods [13,14]. These data also lend support to previous studies indicating that rod photoreception is newer than cone photoreception [15], since rhodopsin first arose after 2R [4]. This all has been summarized in recent reviews [13,16,17].

In this work, we performed a thorough analysis of the phylogenetic history of the cyclic nucleotide-gated (CNG) cation channels in metazoans. These heterotetrameric channels, located on the cell membrane of the outer segments of vertebrate photoreceptor cells, play an essential role in phototransduction. In darkness high intracellular levels of cGMP keep the CNG channels open, maintaining a $Na^+$ influx that keeps the photoreceptor cells depolarized and $Ca^{2+}$ influx that regulates phototransduction proteins. When light activates the cascade, there is a reduction in cGMP levels and a subsequent hyperpolarization of the photoreceptor cell. In vertebrates, just as cones and rods express different opsins, transducin subunits [6,18] and PDE6 subunits [7], different CNG subunits are used to form the functional channels: cones use three CNGA3 subunits and one CNGB3 subunit (previously thought to be two of each of the cone-specific subunits) while rods use three CNGA1 subunits and one CNGB1 subunit [19]. The formation of the functional channel is facilitated by a C-terminal leucine zipper domain of cyclic nucleotide-gated channels (CLZ) of the CNGA subunits [20].

Besides being used in the visual system, CNG channels are also involved in olfaction, with olfactory receptor heterotetramers consisting of two CNGA2 subunits, one CNGA4 subunit and one CNGB1 subunit, where CNGA4 mainly has a modulatory role [21,22]. For a summary of CNG structure and function see [23].

Putative homologs of vertebrate CNG genes have been identified in prokaryotes [24] suggesting an origin very early in evolution. The signaling cascade utilizing an opsin, G-protein of the $G\alpha_{i/t}$ family, PDE and CNG channel is called a ciliary phototransduction and previous studies have suggested that it preceded the appearance of bilaterians [25]. In non-vertebrate metazoans CNG channels are used in the phototransduction signal cascades together with opsins and arrestins. Examples of this signal cascade can be found in the neural cells that regulate cnidocytes in the hydra (*Hydra vulgaris (magnipapillata)*) [26]; in the eyes, antennae and brain of *Drosophila melanogaster* [27]; in the photosensitive neural subtypes in the anterior nervous system in the annelid *Platynereis dumerilii* [28]; and in the ciliated anterior apical trunk epidermal neurons expressing GnRH (that are likely chemosensory) in the larval stage of the tunicate *Ciona intestinalis* [29]. It has also been speculated that the amphioxus frontal eye uses a CNG channel in its photoreceptors [14], but no expression data have been published.

Extant vertebrates typically have six genes encoding CNGs, excluding lineage-specific duplications or losses: *CNGA1-4*, *CNGB1* and *CNGB3* (*CNGB2* has been renamed *CNGA4*). These genes belong to two distinct and distantly related subfamilies, *CNGA* and *CNGB*, within the CNG family [2,3]. When our initial evolutionary studies were performed, only human genomic information was available for analyses of conserved synteny [2]. Consequently, it was not possible to attribute the CNG gene duplications to 2R with certainty. This was probably due to translocations that appeared to conceal the paralogon, i.e., a quartet of related chromosomal regions [2,3]. Later studies, based on phylogeny of the CNG gene family, suggested that a local duplication of an ancestral *CNGA1/2/3* gene took place before the protostome-deuterostome split, resulting in *CNGA4*. Later the 2R duplications resulted in *CNGA1-3* in extant vertebrates. Duplication of the ancestral CNGB gene in 2R and subsequent losses resulted in *CNGB1* and *CNGB3* (see Lamb [13] and references therein). However, these later studies lacked detailed analysis of conserved synteny for neighbors of all the CNG genes that could resolve any chromosomal rearrangements that have previously shrouded the picture.

Knowledge about the evolutionary history of the genes involved in vision provides a better understanding of the origin and development of complex organs in vertebrates. For example, investigations into the cause of the high prevalence of achromatopsia in the population residing on the atoll of Pingelap in Micronesia (about 5%) resulted in the identification of the causative mutation (in a locus named *ACHM3*) and the identification of the cone specific *CNGB3* [30,31]. This atoll and its residents were described by the famous neurologist Oliver Sacks in his book "The island of the colorblind" [32]. Up to 90% of achromatopsia cases are due to mutations in *CNGA3* or *CNGB3* with approximately 100 mutations described in each of these genes [33,34].

Here, through extensive searches in metazoans we identified four previously unknown CNG genes that were present in the ancestor of bilaterians and cnidarians. Additionally, we took advantage of a broad repertoire of vertebrate species with high-coverage chromosome scale genome assemblies for detailed phylogenetic and conserved synteny analyses to see if we could disentangle the evolutionary history of the vertebrate CNG genes. We observed that conservation of synteny in vertebrates has been disrupted by chromosomal rearrangements, but in different ways in the different evolutionary lineages. By combining data from these lineages, we can now conclude that the cone-specific and rod-specific CNGA and CNGB subunit genes did indeed arise in the basal vertebrate WGDs, re-expanding the gene counts to numbers like the bilaterian and cnidarian ancestor. Additionally, by also investigating the CNG sequence repertoire in actinopterygians, a lineage having a wide range of visual opsin gene repertoires due to life in diverse visual environments [35], we can conclude that most surviving gene duplicates of the CNG subfamilies are the result of WGDs, while local duplications have been kept to a minimum. Taken together, our results provide further support that the vertebrate WGD events have made major contributions to the molecular machinery used for vision in cones and rods.

## Results

### The early origin of the CNGA and CNGB genes reveals unexpected multiplicity

The vertebrate genes encoding the CNG subunits form a family that has previously been subdivided into two subfamilies, *CNGA* and *CNGB*, originating in Bilateria before the protostome-deuterostome divergence [2,3,14]. Here, we attempted to date this duplication by performing extensive searches in eukaryotes. Our analyses suggest that there was a CNGA and a CNGB type gene before the radiation of metazoans, as shown by our BLASTP searches in non-

metazoan eukaryotes where we identified many sequences that have human CNGA and CNGB sequences as best reciprocal BLASTP hits (2252 versus 73 respectively). However, none of the putative CNGA sequences outside of metazoans have the CLZ domain when running hmmscan aginst PfamA with standard settings. When using the criteria of conserved domain architecture, we can date the duplication of *CNGA* and *CNGB* to at least the early metazoan ancestors, since we identified sequences with the same domain architecture as human CNGA, as well as human CNGB in the sponge, *Amphimedon queenslandica* (Figs 1–3).

We identified five CNGA type gene clades and we propose a naming for these five clades:—CNGA (named upon traditional CNGA sequences) in which most genes have a CLZ domain and containing representative sequences of vertebrates and sponges;—CNGC present in both Ctenophore, Cnidaria and some Bilateria clades;—CNGD which is found in most of metazoans except in sponges and placozoans;—CNGE which is only found in cnidarians;—CNGF which is present in protostomes and cnidarians but which have been lost in deuterostomes (Figs 1 and 3). Only 29 species of non-cnidarians have more than the four identified CNGA type genes. Out of these, with some exceptions, the majority are either rotifers (seven species) or chelicerids (five species). The animals with the largest number of CNGA type genes are tardigrades with nine genes.

Previous studies have suggested that the duplication that gave rise to *CNGA4* and the ancestral *CNGA1/2/3* occurred before the protostome-deuterostome split [13,14,36]. However, our phylogenetic analyses with a broader representation of taxa clearly place the local gene duplication that generated *CNGA4* and *CNGA1/2/3* in the vertebrate predecessor after it diverged from the tunicate ancestor (Figs 1 and 4). Tunicates, lancelets, and echinoderms all have *CNGA* sequence(s) that with high support branch off earlier than the vertebrate sequences (Figs 1 and 4). Most likely the previous results were due to the substitution model used and/or the higher evolutionary rate of *CNGA4* as compared to *CNGA1-3*, as well as the tunicate *CNGA* (Fig 4).

We were only able to identify one original member of the *CNGB* family (Fig 2) and most species only have a single gene: out of 271 non-vertebrate metazoans, only 44 have more than one and most of these species have two. Ten species have three or four CNGB genes. Again, tardigrades have the largest number of sequences (up to four) and rotifers and chelicerids are again among the species with the largest number of genes.

## The vertebrate CNGA genes

The phylogenetic tree of the CNGA amino acid sequences shows three well supported bony vertebrate clades; one for *CNGA1*, one for *CNGA2* and one for *CNGA4* (Fig 4). The *CNGA3* sequences do not form a single well-supported clade, but rather several where three separate clades have high support: one of actinopterygian fishes, one of amniotes, and one of cartilaginous fish (Fig 4). This gene is syntenic with *CNGA4* in the spotted gar, chicken, and small-spotted catshark genomes (5 Mbp, 62 Mbp and 22.7 Mbp apart, respectively) supporting its identity as *CNGA3* and suggesting that a local duplication that took place before 1R and 2R generated the ancestral gene pair. The split between the *CNGA1-3* sequences and *CNGA4* is well supported indicating that *CNGA1-3* resulted from duplications after the *CNGA4* ancestor had branched off, most likely in 1R and 2R (Fig 4). We identified two putative orthologous sequences to gnathostome *CNGA4* among lamprey proteins in the NCBI non-redundant protein database (Fig 4). We queried agnathan genomes in the NCBI database using the human *CNGA4* amino acid sequence and found that *CNGA4* are not located on the same chromosome as any putative *CNGA1-3* in any of the lamprey or hagfish genomes (S2 Table). The far eastern brook lamprey (*Lethenteron reissneri*) genome, however, has two CNGA genes located

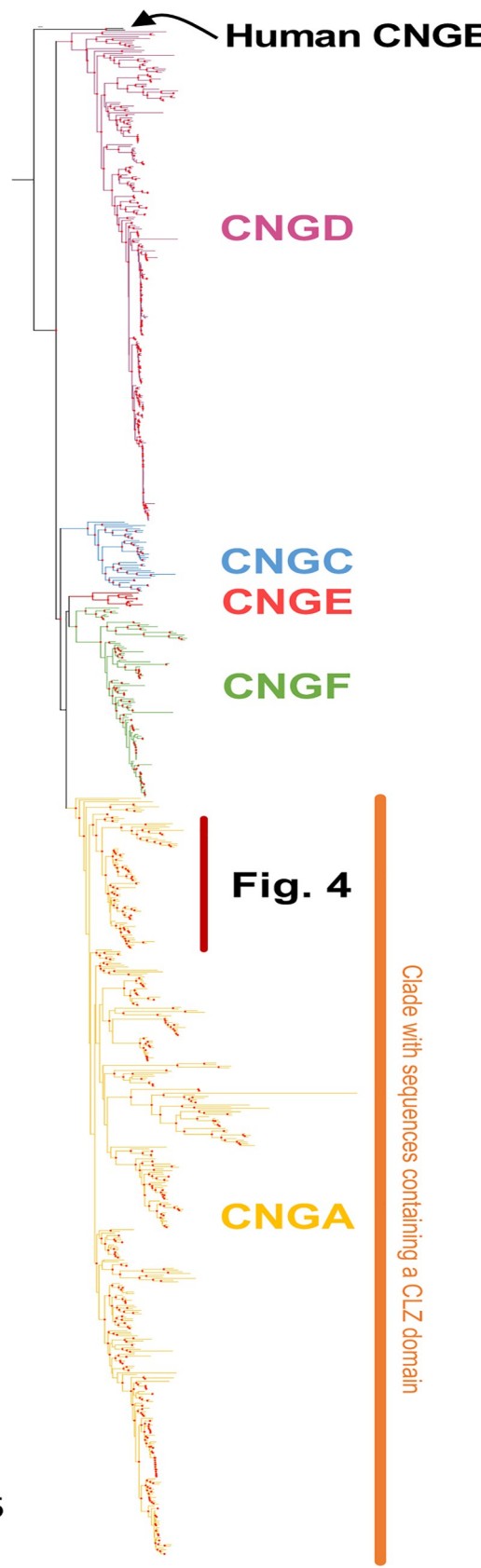

**Fig 1. Phylogenetic analysis of all identified metazoan CNGA and CNGA-like sequences.** Colors represent well supported metazoan subfamilies of the CNGA subtype and its most closely related subtypes as identified in this analysis. The colored text represents the proposed name of each novel subtype. The CLZ domain, important for trimerization of the CNGA subunits, was identified only among the classical CNGA genes. The tree was constructed using IQ-Tree version 1.6.1 with 10,000 ultra-fast bootstrap replicates. The tree shown is the consensus tree, nodes are considered strong when they had a support ≥ 90%. Well supported nodes have been labelled with a filled red circle. The phylogenetic trees were rooted with the human HCN1-4 sequences and human CNGB sequences was included as reference respectively.

on the same chromosome, but they are probably the result of a lineage specific duplication or an assembly error due to the identical amino acid sequence (S2 Table). Analysis of actinopterygian CNG genes is described in a separate section.

## The vertebrate CNGB genes

The phylogenetic tree of the amino acid sequences of the *CNGB* subfamily forms a well-supported clade for the vertebrate *CNGB1* sequences (Fig 5). The same goes for the *CNGB3* clade which has high support (Fig 5). Two lamprey sequences appear to be *CNGB1* while three other sequences branch of basally to all gnathostome GNB sequences (Fig 5). Analysis of actinopterygian CNG genes is described in a separate section.

## Evolution of CNG genes in actinopterygian fish

To investigate the CNG gene repertoire further in a group of vertebrates with a rich and dynamic opsin gene repertoire, we complemented our analysis by performing reciprocal BLASTP searches against the NCBI RefSeq protein database restricted to actinopterygian fish. The identified sequences were aligned and subjected to phylogenetic analysis. From these data it appears as if most duplicates are the result of WGDs. We observed that non-teleost actinopterygian fish have four CNGA genes, except for sturgeons and paddlefishes that have seven genes (Fig 6A). In general, teleost fish have seven CNGA genes, with exceptions e.g. cyprinids and salmonids have experienced extra lineage specific WGDs (Fig 6A). Teleost *CNGA1*, *CNGA2* and *CNGA3* have duplicates located on separate chromosomes, which is indicative of duplication in 3R. Further investigation of the chromosomal locations of the genes in actinopterygian fishes, with chromosome level assemblies, indicated four local/tandem duplications in *CNGA3* in the green spotted pufferfish (Fig 4), in *CNGA1* of the jewelled blenny (*Salarias fasciatus*), *CNGA2* of the mummichog (*Fundulus heteroclitus*), *CNGA3* in the sterlet (*Acipenser ruthenus*) and *CNGA4* in the goldfish (*Carassius auratus*) (S2 Table). We observed that most species have between three and four CNGB genes with some exceptions. Like CNGA genes, some cyprinids and salmonids have extra duplicates (Fig 6A). Notably, reedfish and gray bichir have a lineage specific local duplication of *CNGB1* (Figs 5 and 6A). Investigation of the chromosomal locations of the CNGB genes in teleost fish revealed that most putative local duplications are for *CNGB3*, due to the proximity of the genes (~22 kbp in zebrafish, S1 and S2 Tables). Similarly, we found that the Asian arowana genome (*Sclerophages formosus*), a teleost species belonging to a group (bonytoungues) that diverged early from most other teleost species, has two copies of *CNGB3* located close to one another on the same chromosome (chromosome 19, ~16 kbp apart in the fSclFor1.1 assembly, Ensembl 102) (S18B Fig in S1 File, S2 Table). Indeed, these two *CNGB3* duplicates seem to be present in many teleosts genomes available in Ensembl 102. The presence of only one *CNGB3* in spotted gar and reedfish lends further support for a local duplication in the teleost ancestor (Figs 5, 6A and 6C). Taken together, this indicates that the duplication occurred early after 3R. Chromosome level assemblies also reveal that only three of these teleost fish species have putative local duplicates for *CNGB1*, namely

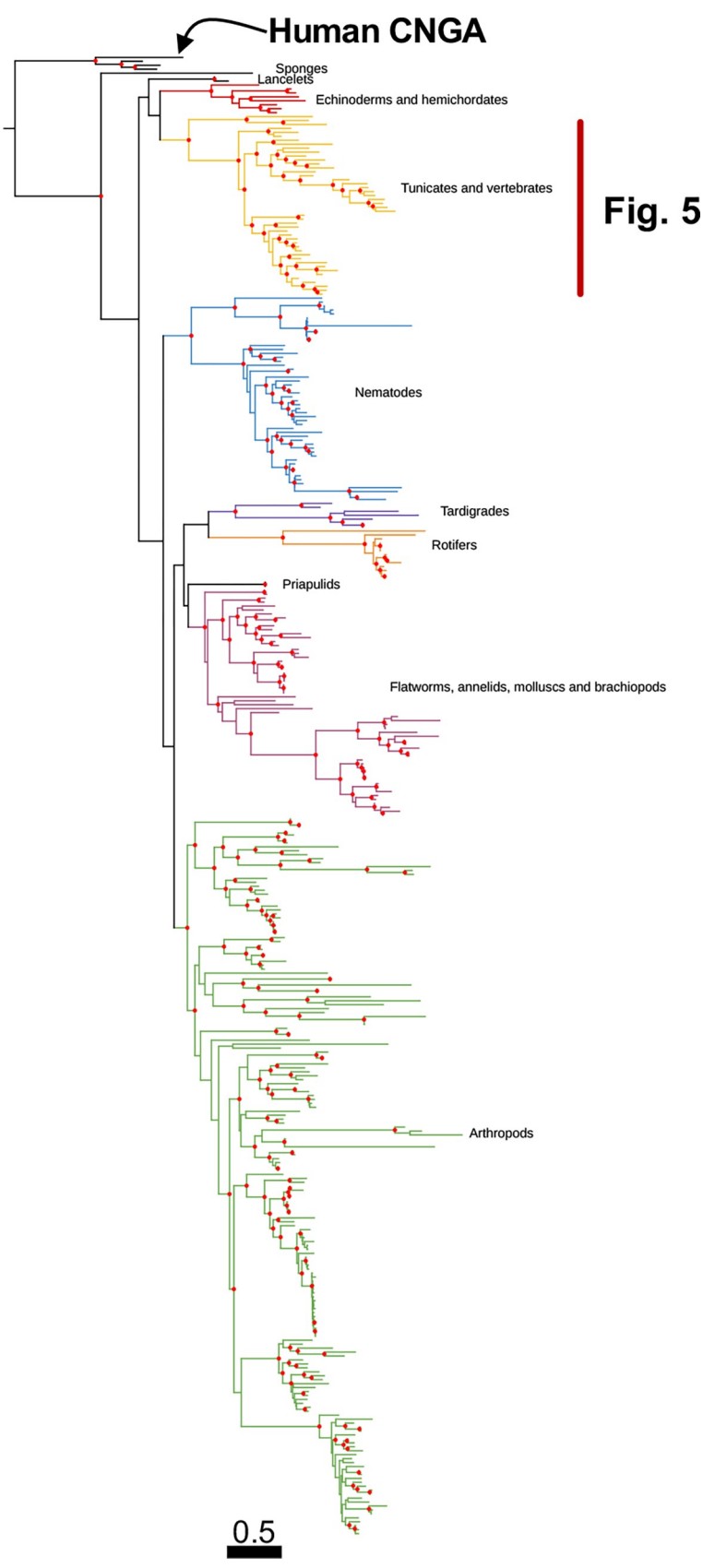

**Fig 2. Phylogenetic analysis of all identified metazoan CNGB type sequences.** Colors represent strongly supported clades labelled with metazoan groups that are included in each clade. The tree was constructed using IQ-Tree version 1.6.1 with 10,000 aLRT and ultra-fast bootstrap replicates. Nodes are considered strong when they have an aLRT supports ≥ 80% and an ultra-fast bootstrap support ≥ 95%. Well supported nodes have been labelled with a filled red circle. The tree was rooted with the human HCN1-4 sequences and human CNGA sequences were included as reference.

the Siamese fighting fish (*Betta splendens*), the lake whitefish (*Coregonus clupeaformis*) and the Sumatra barb (*Puntigrus tetrazona*) (Fig 6A, S2 Table). The Japanese pufferfish genome contains eight CNGB genes, most being located on the same chromosome (chromosome 10). This is most likely due to a lineage specific local duplication or is an artefact of assembly error (Fig 6A, S2 Table).

## Comparative synteny analysis of the chromosomal regions harboring the CNGA genes

In the selection of neighboring gene families, we identified nine gene families with members on the three *CNGA*-carrying chromosomes in spotted gar. Out of these nine, two gene families encode the GABA$_A$ α and β subunits. These gene families have been analyzed independently by our group and will be presented in a separate manuscript describing the complex evolution of all GABA$_A$ receptor subtypes in vertebrates (Haines *et al.*, in preparation). The phylogenetic trees of the remaining seven neighboring gene families are shown in S1-S7 Figs in S1 File. Of these seven families, *KCTD*, *BMX*, *RAB33*, *ELF* and *NIPA* have a topology that is consistent with an expansion in the timeframe of 2R. *EDNR* and *PCDH* did not have non-chordate sequences annotated that could be used as outgroup. This means that although the topology is compatible with an expansion in 2R it is difficult to date their duplications relative to the 2R events.

After mapping neighboring gene family members to chromosomes in six vertebrate genomes with chromosome level assemblies, we observed several instances of gene translocations in human compared to the other gnathostomes included in the analysis (Fig 6B). In particular, one of the four chromosomes (with the genes marked as green boxes) in this paralogon that has undergone rearrangements in human, dispersing the genes to four different chromosomes (2, 11, 13 and 15), whereas they are neatly syntenic in chicken, reedfish, spotted gar and small-spotted catshark. Somewhat confusingly, one *BMX* family member has been translocated to human chromosome X that already has a member of this family. Although the CNGA3 genes in the phylogenetic analysis of the *CNGA* family did not form a single clade, the comparison of synteny provides evidence in favor of these sequences being orthologs (Figs 4 and 6B). When comparing the conservation of synteny between the spotted gar and reedfish (representing an even earlier diverging actinopterygian lineage), we observe a high level of conservation (Fig 6B). Comparison of zebrafish with spotted gar and reedfish shows several duplications consistent with 3R, but also many translocations compared to the two non-teleost genomes, most likely following 3R, as has been observed in different lineages after WGD events by us and others [11,37–39].

The small-spotted catshark displays strong conservation of synteny to the spotted gar and reedfish, indicating that this is the ancestral gnathostome organization (Fig 6B).

## Comparative synteny analysis of the chromosomal regions harboring the CNGB genes

We identified 11 neighboring gene families for the CNGB genes that have members on both chromosomes with CNGB genes in spotted gar. Out of these families one (*SPG7/AFG3*

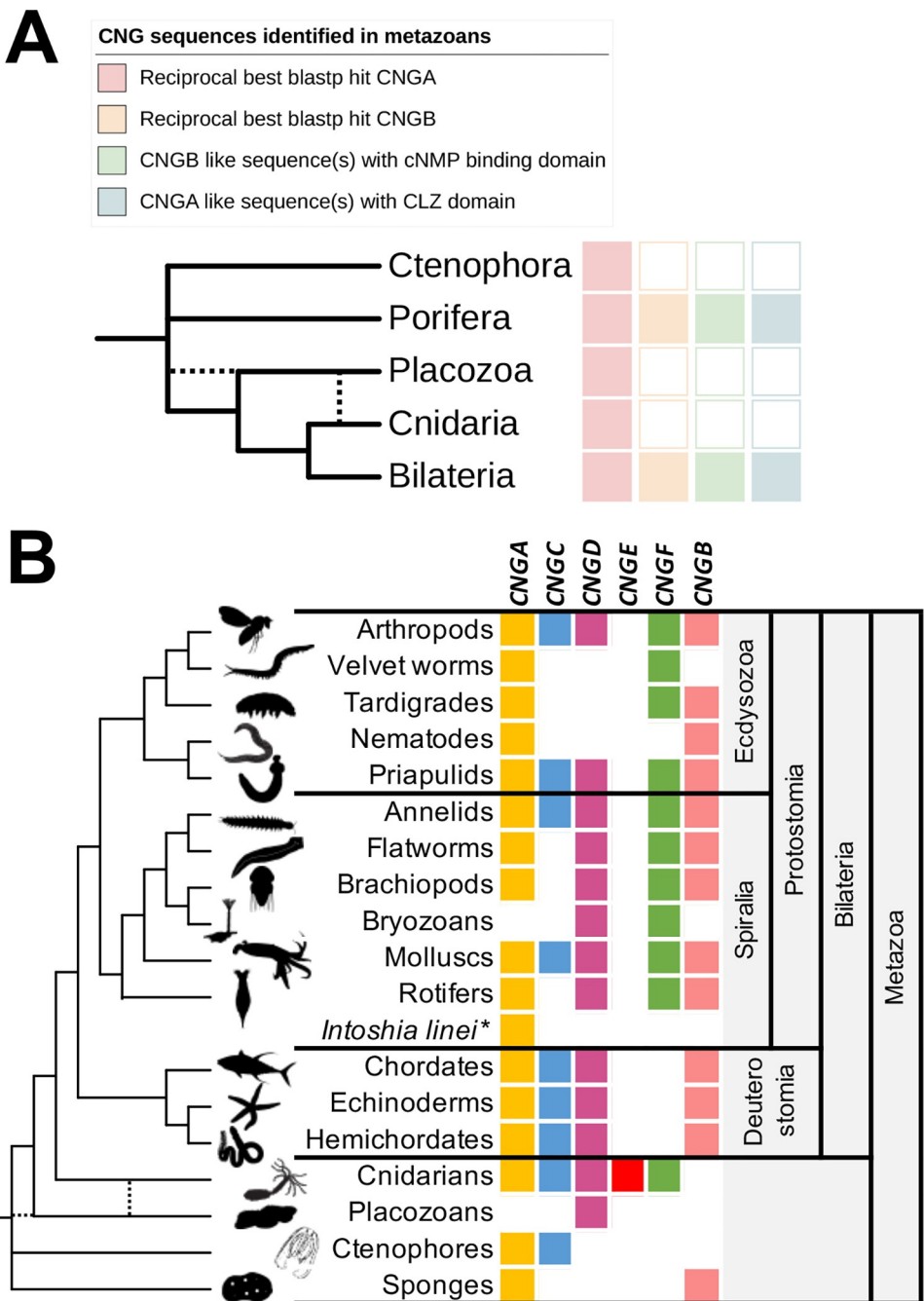

**Fig 3. CNGA and CNGB type genes in metazoans.** A) Summary of BLASTP searches for CNG genes in metazoans after PfamA screening for CNG channel domains. The results show that only animal groups that have CNGA genes with a CLZ domain also have genes of the CNGB type. Dotted lines represent alternative branching for the placozoa lineage. B) Presence or absence of genes of the CNGA and the CNGA-like subtypes in the different animal groups included in the analysis. The results show that there have been several lineage specific losses within protostomes and deuterostomes. The Bilaterian ancestor most likely had CNGA, CNGC, CNGD and CNGF genes. The ancestor of deuterostomes lost the CNGF gene. The only lineage that has genes of all CNGA-like subtypes is Cnidarians. *Intoshia linei* is labelled with an asterix due to its uncertain position. Silhouettes were retrieved from phylopic.org and are all dedicated to the public domain.

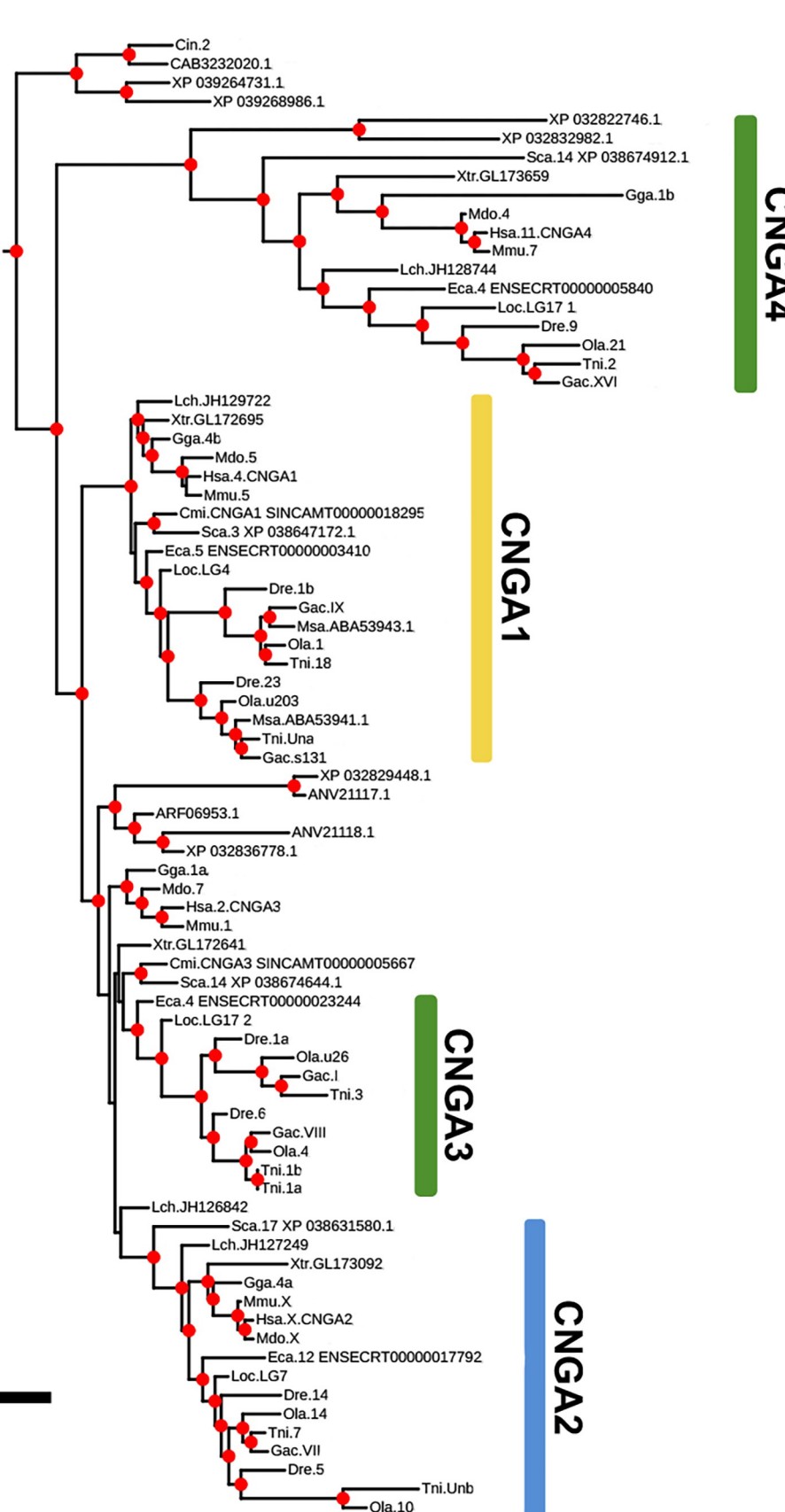

**Fig 4. Enlarged portion of Fig 1 showing the Olfactores clade of the classical CNGA sequences.** Vertebrate clades are colored based on the chromosome of the closest spotted gar ortholog and corresponds to colors used in Fig 8B; CNGA1 – yellow, CNGA2 blue, CNGA4 and CNGA3 –green.

ENSFM00730001521276) was excluded due to expansion in a time-frame other than 2R. The other 10 families show a duplication history compatible with an expansion in 2R (see S8-S17 Figs in S1 File).

Chromosome blocks with clear conservation between the non-teleost actinopterygian fishes spotted gar and reedfish were identified analogously with the CNGA paralogon (Fig 6C). Also,

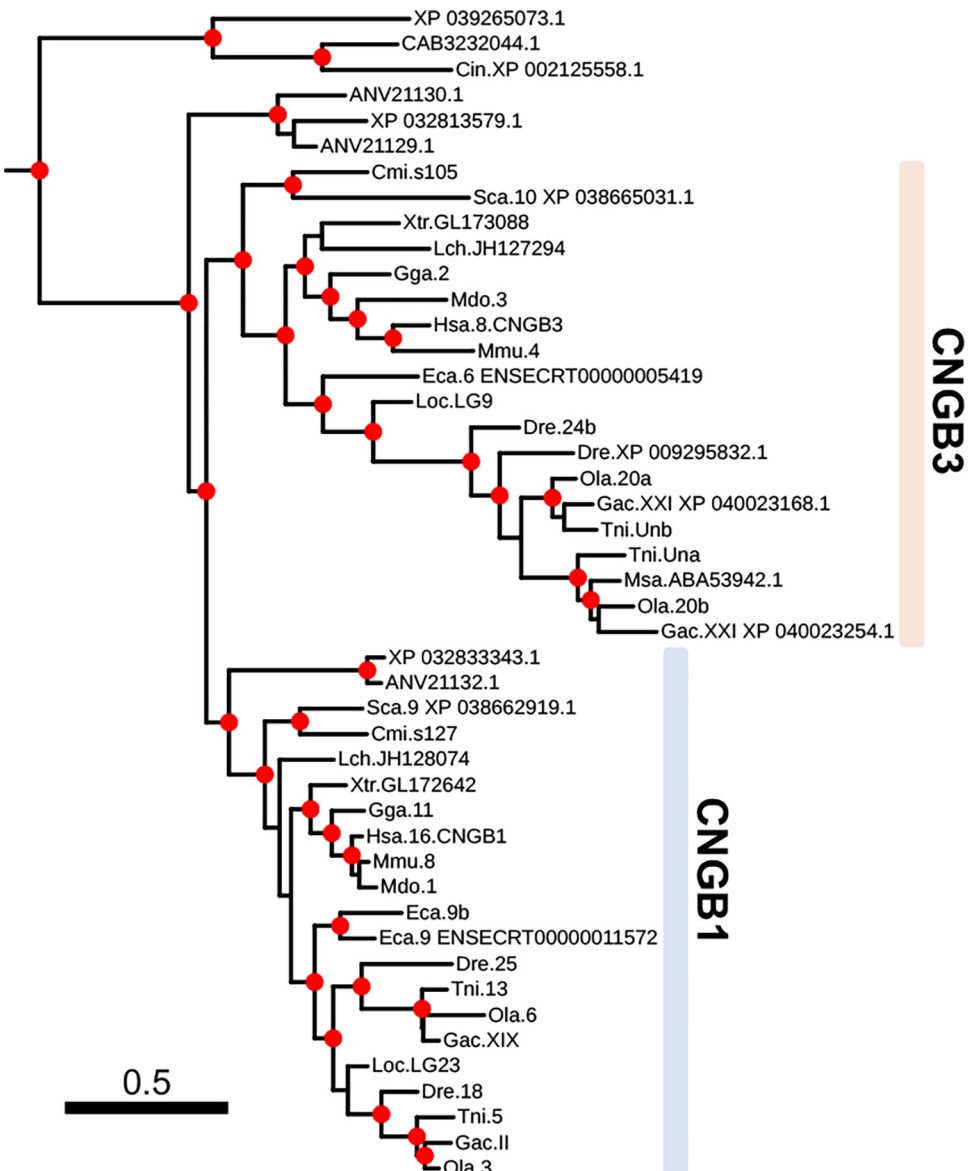

**Fig 5. Enlarged portion of Fig 2 showing the Olfactores clade of CNGB sequences.** The gnathostome sequences clearly have been subdivided into two clades, CNGB1 and CNGB3. Vertebrate clades are colored based on the chromosome of the closest spotted gar ortholog and corresponds to colors used Fig 8C; CNGB1 –light blue, CNGB3 –light pink.

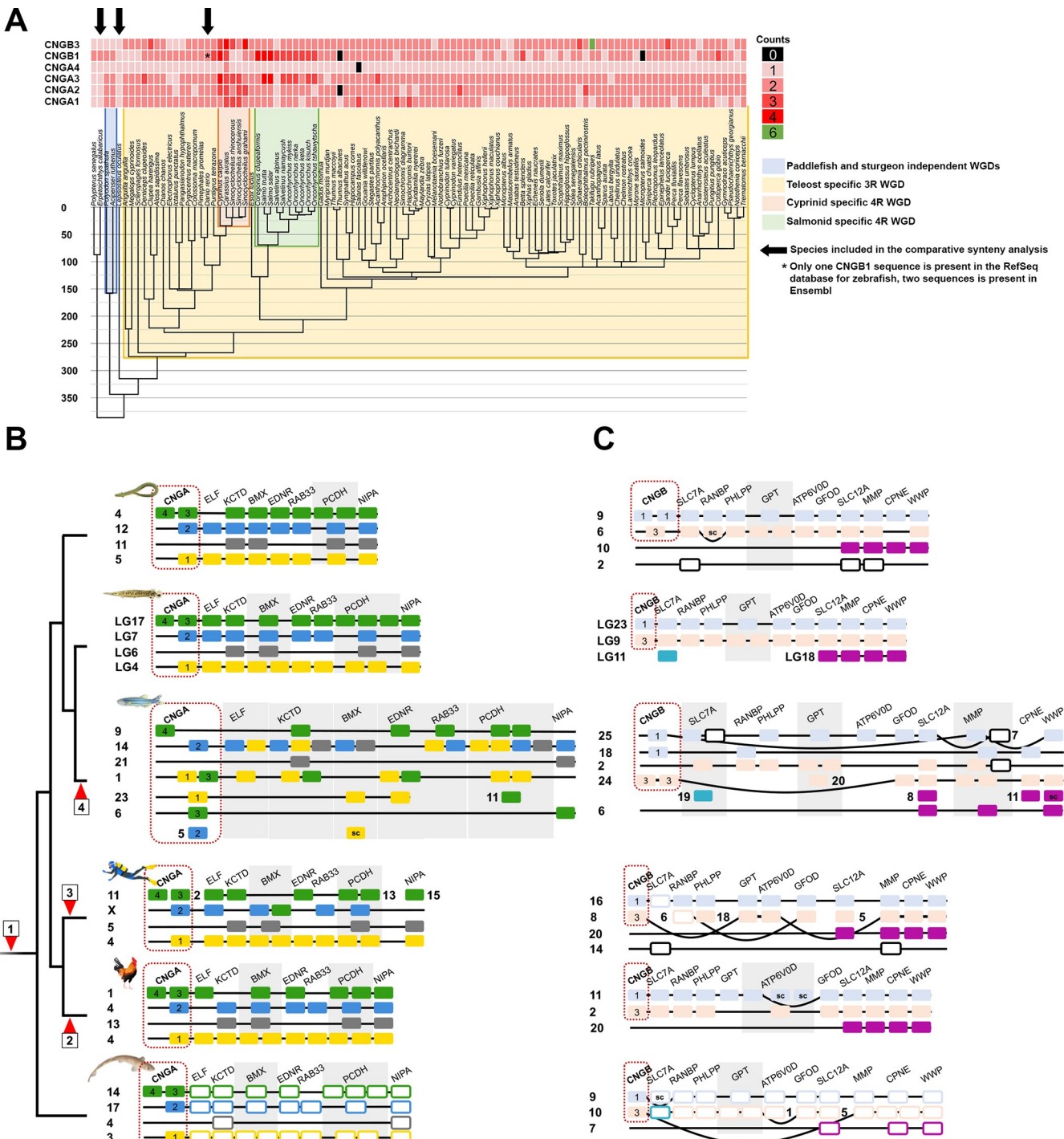

**Fig 6. CNG sequence counts mapped to a time-calibrated phylogeny of ray-finned fish and comparative synteny analyses of CNGA and CNGB gene regions in select vertebrates. A)** A time-calibrated phylogeny retrieved from timetree.org of actinopterygian fishes and the CNGA and CNGB gene counts as a heatmap. Lineages marked with a colored background have experienced extra whole genome duplications. B) Chromosomal neighborhoods of CNGA genes in the spotted gar genome used in the phylogenetic analyses, and their orthologs, and their neighboring gene family members, in reedfish, zebrafish, human, chicken and small-spotted catshark. C) Chromosomal neighborhoods of CNGB genes in the spotted gar genome used in the phylogenetic analyses and their orthologs, as well as their neighboring gene family members, in reedfish, zebrafish, human, chicken and small-spotted catshark. Order of genes has been reshuffled to highlight similarities between the linkage groups within the paralogon and between species. The phylogenetic trees of the neighboring gene families are shown in S1-S17 Figs in S1 File. Red arrow heads represent: 1) 2R WGDs. 2) Chromosomal fusion in the chicken lineage; 3) rearrangements in the mammalian lineage; and 4) 3R WGD in teleost fish. Spotted gar chromosomes are colored as they are in Fig 1B and 1D, while the other animals' orthologs are colored the same as their spotted gar ortholog–see Figs 6 and 7. Boxes for neighboring gene families in the small-spotted catshark have been left white with colored edges since they are not included in phylogenetic analyses. Numbers in gene boxes indicate which CNGA or CNGB gene the box represent.

like for the *CNGA* paralogon, we observed evidence for 3R followed by major rearrangements in zebrafish when compared to the other two actinopterygian fish species (Fig 6C). Comparison of synteny between spotted gar, human and chicken indicated rearrangements in mammals, although not as extensive as for *CNGA* described above (Fig 6C). It is nevertheless clear that the CNGB genes are located within blocks of conserved synteny that provide strong evidence for orthology of the *CNGB3* gene in these species.

We also investigated the location of the best ortholog matches for each neighboring gene family member in the small-spotted catshark genome. Thereby we observed extensive conservation of synteny with spotted gar and reedfish. However, in contrast to the *CNGA* paralogon, there seem to have been some rearrangements in the small-spotted catshark genome in the *CNGB* paralogon (Fig 6C).

Gene loss has been more extensive in the *CNGB* paralogon than in the *CNGA* paralogon, and only reedfish, zebrafish and human (and other mammals) have gene families with a fourth paralogon member (Fig 6C): only a single gene family is a quartet in human and zebrafish (*MMP*) and two families are quartets in reedfish (*MMP* and *SLC12A*). Also, *SLC7A* supports a fourth chromosome member of the paralogon, but this family has lost two 2R paralogs.

## Expression of CNG genes outside of vertebrates

To investigate the possible composition of CNG channels in non-bilaterian metazoans we queried scRNA-seq datasets from a ctenophore, a sponge, a placozoan and a cnidarian [40,41]. When plotting the expression of *CNGA* and *CNGC* identified in the ctenophore *Mnemiopsis leydi*, we observed that the two genes have different expression patterns in mostly non-overlapping metacells (groups of highly similar single cells) (Fig 7A). *CNGA* has a much broader expression in many metacells that have among others been identified as sensory, neuronal, or part of the subepithelial nerve net [42]. *CNGC* on the other hand has its highest expression in the metacells identified as digestive in the original publication. Similarly, in the adult sponge, *Amphimedon queenslandica*, *CNGA* has broad expression in many metacells of different types while *CNGB* has strong expression in bactericidal, aspcinzin and pinacocyte cells (Fig 7B). In the larval sponge, both genes are co-expressed in several metacell subtypes (Fig 7B'). The single gene identified in the placozoan *Trichoplax adherens*, *CNGD*, is expressed in epithelial, digestive and lipophile metacell types (Fig 7C). Finally, three of the five CNG genes identified in the cnidarian *Nematostella vectensis*, the non-bilaterian with the most CNG genes, show overlapping expression in similar cell types in the adult animal with the highest expression of *CNGD*, *CNGE*, and *CNGF* in neuronal, secretory gland, gastrodermis and digestive filament metacells (Fig 7D). *CNGA*, however, seem to be mostly expressed in digestive filaments and *CNGC* has very low expression overall (Fig 7D). In the larval *Nematostella vectensis* only three genes are expressed, *CNGD*, *CNGE* and *CNGF*. Out of these gens *CNGE* has the highest expression, which is strongest in a in neuronal metacell (Fig 7D').

By plotting the expression of the identified *Ciona intestinalis* CNG genes in a scRNA-seq dataset of neural cell types during development [39] we observed a similar decoupling of expression between CNGA and CNGB genes as in the adult sponge, but where *CNGD* appeared to take the role of *CNGB* being co-expressed with CNGA in some cells (Fig 7E). There are two cell types that express *CNGB* and *CNGA* at high levels, coronet cells and Rx + aSV cells, of which the latter are photoreceptor cell progenitors. They are also co-expressed in photoreceptor cells (Opsin1+ STUM+ aSV cells and Opsin1+ PTPRB+ aSV cells) but at low levels. Finally, like in the sponge, some *Ciona intestinalis* cell types only express *CNGB* (Fig 7E).

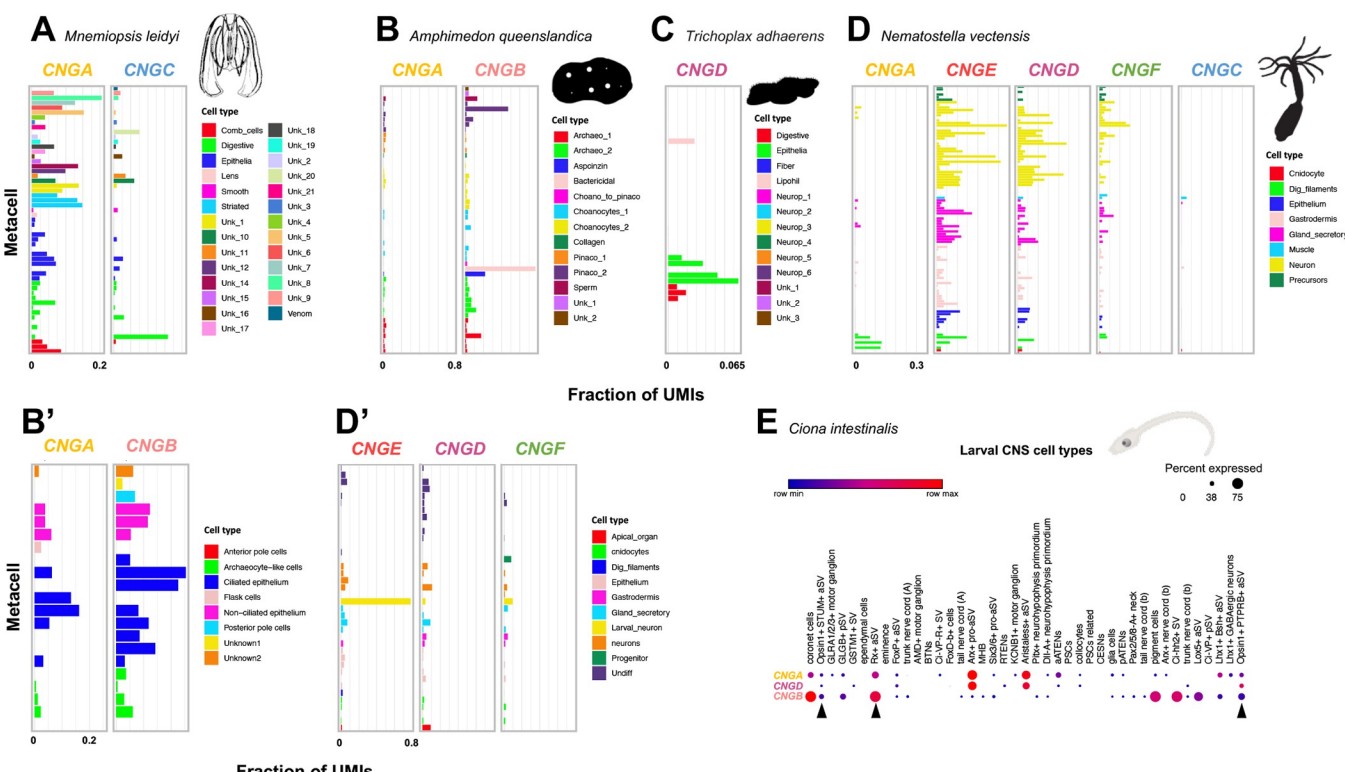

**Fig 7. Expression of CNG genes in non-bilaterian metazoans and in the tunicate *Ciona intestinalis*.** A) Expression of *CNGA* and *CNGC* in ctenophore metacells shows differential expression across cell types. B) Expression of *CNGA* and *CNGB* in sponge adult metacells show, like in the ctenophore, differential expression of both genes. B') Co-expression of both genes in the larval sponge. C) Expression of the single CNGD gene in the placozoan metacells. D) Expression of the five CNG genes in the adult cnidarian *Nematostella vectensis* show overlap in expression in the same metacells for *CNGE*, *CNGD* and *CNGF*, while *CNGA* and *CNGC* have lower expression. *CNGA* has its highest expression in digestive filament metacells. D') In the larval *Nematostella vectensis* only *CNGE*, *CNGD* and *CNGF* are expressed and *CNGE* has the highest expression in larval neurons. E) Expression of the identified CNG genes in the *Ciona intestinalis* larval central nervous system show both overlapping expression of *CNGA* and *CNGB*, and *CNGA* and *CNGD* as well as expression of only *CNGB* in some cells. Black arrows represent photoreceptor cells in the anterior sensory vesicle or their precursors. Silhouettes were retrieved from phylopic.org and are all dedicated to the public domain. Illustration of *Ciona intestinalis* larva was kindly provided by Dr. Daniel Ocampo Daza.

## Discussion

This analysis of the evolution of the CNG genes was sparked by our previous finding that the separate CNG gene subtypes for rods and cones appeared to have arisen in the early vertebrate tetraploidizations [2,3]. However, that conclusion was based largely on investigation of the human genes, and we had noticed that the chromosomal regions containing the CNG genes seemed to have been subjected to rearrangements [3], precluding assignment to specific paralogons. Additionally, the origin of the different CNG genes had not been described in detail outside of vertebrates. Therefore, we first set out to date the origin of the genes encoding CNGA and the CNGB subunits in metazoans, then we employed comparative synteny analyses in a broad range of vertebrates with the spotted gar genome [10] as a reference to resolve the vertebrate genome rearrangements. Lastly, we investigated the gene family expansions in teleost fish, a group of vertebrates that have diverse retinal specializations.

Here we have discovered previously uncharacterized CNG genes in metazoans and propose the names; *CNGC*, *CNGD*, *CNGE* and *CNGF*. These findings suggest that the metazoan common ancestor had a large CNG gene repertoire (Fig 8). Our analyses did not explore whether these are present outside of metazoans, so it remains possible that they are present in non-metazoan eukaryotes as well. Our analyses also show that the ancestor of Olfactores lost *CNGC*

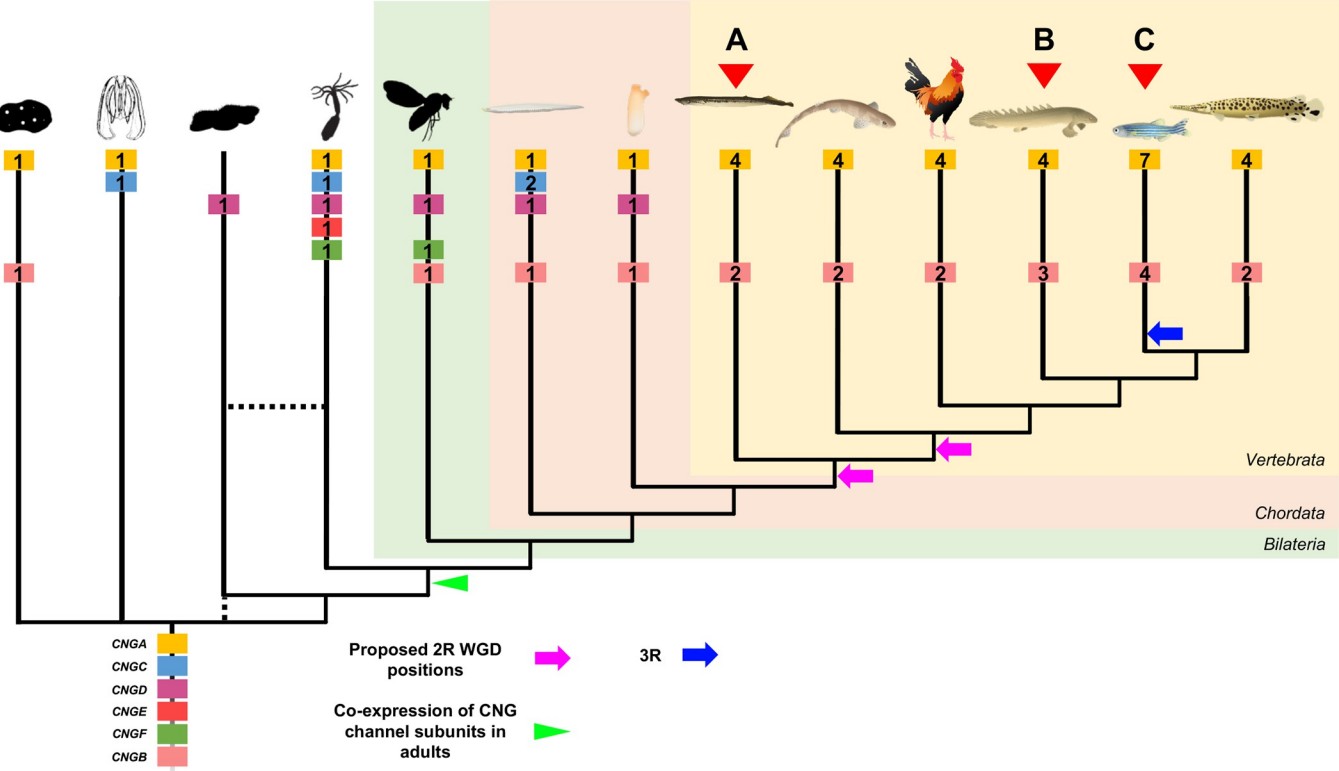

**Fig 8. Evolutionary events leading to the repertoire seen in modern metazoan groups.** Subunit gene repertoire was reduced in the vertebrate ancestor and expanded in the 2R WGDs in the vertebrate lineage either before or after the split between cyclostomes and gnathostomes (purple arrows). Expression of multiple CNG subunits in adult cell types appeared in the ancestor of Cnidaria and Bilateria (green arrowhead). A) Lampreys have an extra duplicate of *CNGA4*. B) The Grey bichir has a local duplicate of *CNGB1*. C) The 3R WGD and a local duplicate of *CNGB3* expanded the number of genes in teleosts further. Numbers in gene boxes represent the number of genes of this subtype in this lineage. The tree is a cladogram and branch lengths does not represent evolutionary distance. Silhouettes were retrieved from phylopic.org and are all dedicated to the public domain. Illustrations of chordates was kindly provided by Dr. Daniel Ocampo Daza.

and *CNGF*, while *CNGE* was lost before the emergence of Bilateria. Later the vertebrate ancestor lost *CNGD*, which has been retained in tunicates, before 1R and 2R (Fig 8). The vertebrate CNGA subfamily is divided into two clades, *CNGA4* and *CNGA1-3* (our results presented here and previous studies [2,3,36]). Our analyses of a broad repertoire of non-vertebrate metazoans show that this duplication took place just before 2R. This conclusion differs from the scenario outlined by Lamb and colleagues, who proposed an even earlier timeframe for the local duplication; before the protostome-deuterostome split (see Fig 5 in [14]; Fig 13 in [13]). We think that the broader selection of species used in this study renders a more parsimonious conclusion. These CNGA genes then duplicated in 1R and 2R (thus quadrupled) whereby the *CNGA1/2/3* ancestor resulted in the present three copies (*CNGA1-3*) after loss of the fourth member. The CNGA4 gene, in contrast, has retained none of the 2R duplicates. Thus, the repertoire in the common ancestor of gnathostomes was four CNGA genes (Fig 8).

Since gene families are affected by differential rates of evolution and selection pressures on their constituent genes, we also investigated in detail the phylogenies of neighboring gene families of *CNGA1-3* and *CNGA4*. We identified a paralogon conserved across several diverse gnathostome species (Fig 6B and S1-S7 Figs in S1 File). We found that the genes orthologous to those located on spotted gar chromosome LG17 had been split across four chromosomes in human (green genes in Fig 6B). We also observed that the orthologs of the genes located on

spotted gar chromosomes LG7 and LG4 had been translocated to chromosome 4 in chicken (blue and yellow genes respectively in Fig 6B). Comparisons of our data with reconstructions of the ancestral chordate karyotype, show that these four paralogous regions belong to paralogon C, F and D in the analysis of Nakatani *et al.* [43] and ancestral group 15 described in the analyses by Sacerdot *et al.* [44] and Lamb (2021) [45]. The most recent publication by Nakatani *et al.*, [46], suggests that the *CNGA* regions originated from proto-vertebrate chromosomes 4, 6, 7, 8, 9 and 11 [46]. Thus, these studies show extensive rearrangements in this region after 2R, further clarifying why it has been difficult to disentangle the evolution of these genes before the present study.

Upon searching for CNGB genes, we only found orthologs of the vertebrate genes in lineages that also have CNGA genes with a CLZ domain (Fig 3A). Previous studies have shown that CNGA subunits first form a heterotrimer, facilitated by the CLZ domain, that later bind to a CNGB subunit to form a functional heterotetrametric channel [20]. If the CLZ domain is removed from CNGA1, the channels combine in an uncontrolled manner with an excess of CNGB1 subunits resulting in a dominant negative effect by forming channels that are not as efficiently expressed in the cell membrane [20]. We therefore postulate that there is a constraint that makes it unfavorable to express CNGB in the same cells as a CNGA subunit without the CLZ domain and that any of the other subtypes of CNGA-type CNG genes take the role of CNGB in these cases.

In vertebrates the *CNGB* subfamily consists of two genes [2,3,14] as confirmed by the analyses described here (Fig 5). The duplication resulted from 1R or 2R as shown by our analyses of conserved synteny and paralogon (Fig 6C). We previously reported that these genes appeared to share evolutionary history with the paralogon of the four opioid receptor genes, which were shown to have duplicated in 2R [2,3,47,48]. This is the paralogon that was named B in the report by Nakatani *et al.* [43] and much later named ancestral linkage group 10 in the supplementary data tables of Sacerdot *et al.* [44] and Lamb (2021) [45]. According to the most recent publication by Nakatani *et al.*, [46], the *CNGB* regions were suggested to have originated from proto-vertebrate chromosome 2 and 3 [46]. Although the *CNGB* subfamily only has two members, we found neighboring gene families on three chromosomes and a putative fragmented fourth member (Fig 6C). Interestingly, the regions corresponding to spotted gar LG9 in smallspotted catshark and human have been independently rearranged providing further clues as to why this was difficult to resolve previously (Fig 6C).

As the visual system of actinopterygians had often experienced extensive duplications of visual opsins [49], we decided to investigate the CNG gene content in many of these species. This revealed a clear pattern where most retained gene duplicates in this lineage are from WGD events: either 1R or 2R, the different sturgeon-specific WGDs, teleost-specific WGD (3R), salmonid-specific 4R WGD and the carp-specific 4R WGD. We found that *CNGA1*, *CNGA2*, *CNGA3* and *CNGB1* all most likely acquired duplicates from the teleost-specific 3R WGD (Figs 4, 5 and 6A and S18A Fig in S1 File). Duplicates of *CNGB3* in teleost fish, on the other hand, are due to a local duplication in the teleost ancestor. In the investigated species, with chromosome level assemblies, nine had putative lineage specific local duplicates that were not the *CNGB3* duplication (Fig 6A and S2 Table). These observations are in strong contrast to observations of gene duplications of the visual opsins, where most post-2R duplications are due to tandem duplications with only a few cases of retained 3R duplicates [49]. A possible explanation is that four subunits are required for each CNG channel and if one gene is duplicated more than the others, the proportions become skewed, while more copies of the visual opsins, allow for a wider spectral range. Interestingly, the only CNG gene that has not retained any extra duplicates (except in goldfish (*Carassius auratus*)) is *CNGA4* (Fig 6A). This may be because it encodes a modulatory subunit, which together with two CNGA2 and one CNGB1

subunit forms the functional channel in olfactory neurons [21]. However, this cannot alone explain why only this gene lacks additional duplicates, since the CNGB subunits also play a modulatory role.

Our investigations into non-bilaterian CNG gene expression reveal that in ctenophores and sponges these genes seem have the highest expression in non-overlapping cell types (Fig 7A and 7B), which indicates that the subunit proteins either form homotetrameric channels or form channels with other distantly related ion channel proteins. This is especially peculiar in the case of *CNGB* in sponges since vertebrate CNGB1 has been shown to be unable to form functional homotetrameric channels [50,51], in contrast to the different CNGA subunits. The CNG genes of the cnidarian, *Nematostella vectensis*, show in several cases overlapping expression (Fig 7D), opening the possibility to form heterotetrameric channels and thus opening the possibility for more functional variation among these channels.

Studies in the cnidarian *Hydra vulgaris (magnipapillata)* have provided evidence for an ancient role in photosensitive neurons [25,52]. A role in phototransduction has also been shown in other non-chordate bilaterians such as the annelid *Platynereis durmelii* and the insect *Drosophila melanogaster* showing that the type of phototransduction cascade of vertebrate cones and rods, i.e., ciliary type, were present before the radiation of metazoans [25,27,28,52]. It is interesting to note that according to our analysis the *Hydra vulgaris* sequence mentioned above is of the *CNGD* type, while both the *Platynereis durmelii* CNG sequences and the *Drosophila melanogaster* sequence, that have been investigated regarding expression, are of the *CNGA* type. Additionally, one CNG gene is enriched in planarian eye transcriptomes [53] and most likely of the *CNGA* type based on similarity by BLASTP (67% identity to this *S. mansoni* sequence included in our tree XP_018650675.1). These data together indicate an ancient role of CNG channels in phototransduction already before the duplications that resulted in the present metazoan repertoire of six subtypes but also that as far as we know, phototransduction cascades utilizing CNG channels in bilaterians most likely use the *CNGA* type.

In the tunicate *Ciona intestinalis*, we observed a similar pattern to the non-bilaterian datasets but also similarities with the expression known from vertebrates. The co-expression of *CNGA* and *CNGD* in certain neuronal cell types might allow for a non-traditional heterotetrameric channel increasing the functional variability. The CNGD subunit might take the role of the CNGB subunit in these cells because it, like CNGB, lacks the CLZ domain and could together with three CNGA subunits, containing a CLZ domain, form a functional heterotetramer. This would agree with the observations by Shuart *et al.*, (2011) which suggest that no CLZ domain and co-expression of CNGA and CNGB leads a dominant negative effect [20]. Like in the sponge, we could observe certain neuronal cell types that only express CNGB at high levels, suggesting a possibility of this subunit forming homotetrameric channels. However, functional studies of the newly identified CNG gene subfamilies must be performed to fully resolve this. The proposed tunicate and vertebrate ancestral CNG channel compositions have been summarized in Fig 9.

The evolutionary analyses presented here reveal that the striped bass (*Morone saxatilis*) CNGA sequences identified by Paillart *et al.* [54] were misidentified as *CNGA1* and *CNGA3*, most likely due to unexpected expression in rods and cones, respectively, and are instead probable *CNGA1* 3R duplicates (Fig 4). We and others have previously demonstrated different types of specializations of phototransduction cascade component gene duplicates in zebrafish (see [6,7,55–57] for examples), later confirmed and further complemented by Ogawa and Corbo (2021) who performed single cell RNA-seq analysis of zebrafish photoreceptor and bipolar cells [58]. This indicates that the unorthodox expression in striped bass is not unlikely. The analysis by Ogawa and Corbo shows that *cnga1a*, *cnga1b* and *cngb1a* are expressed in rods (as would be expected from what is known from tetrapods [23]), while *cngb1b*, *cnga3a*, *cnga3b*,

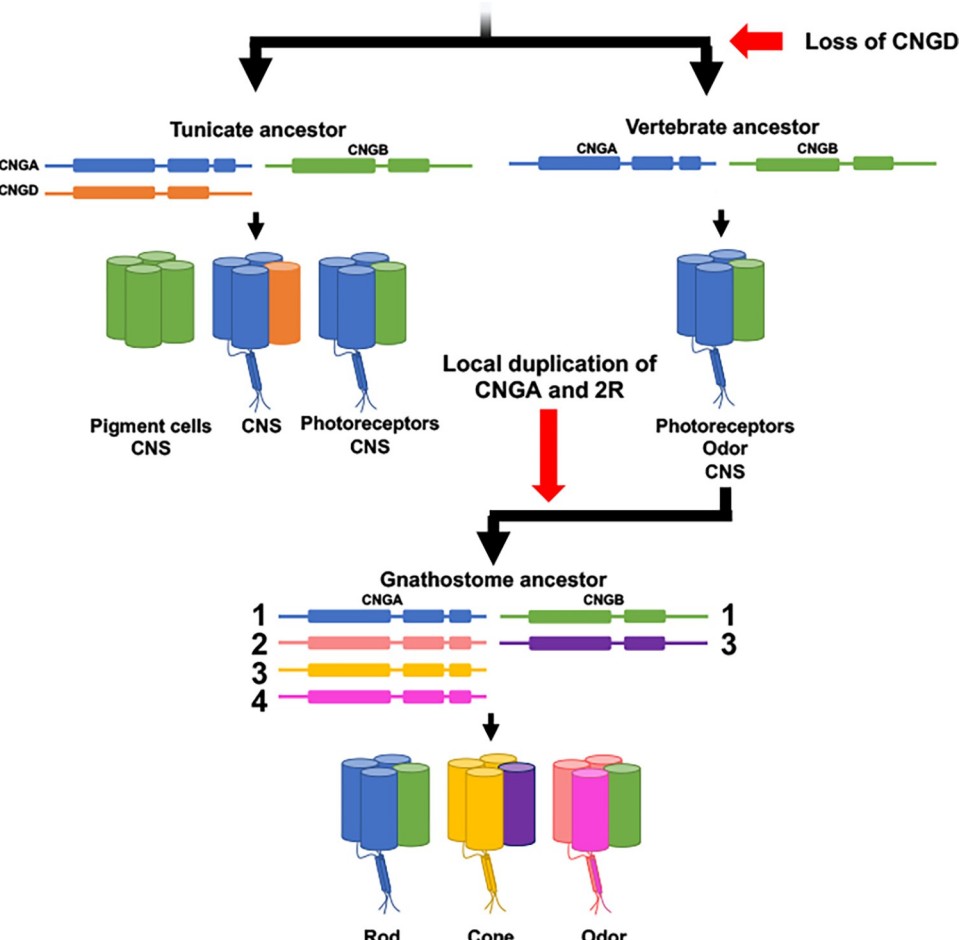

**Fig 9. Evolution and proposed subunit composition in cell types of the tunicate ancestor, vertebrate ancestor and the gnathostome ancestor based on our evolutionary analyses and expression data from *Ciona intestinalis* and human.**

*cngb3.1* and *cngb3.2* are expressed in cones [58]. The expression of *cngb1b* in cones is unexpected and different to the expression of *CNGA1* in cones of striped sea bass [54] indicating a plasticity in cell type assignment of CNG genes in different species of teleost fish. Furthermore, the analysis in zebrafish also shows specialization into different cone subtypes where *cngb1b* and *cngb3.2* are expressed in UV and blue cones; *cnga3a* is expressed in UV, blue, red and green cones; *cnga3b* and *cngb3.1* are mainly expressed in double, red and green cones (see Fig 10 based on clustering data from [58], for a summary). In addition, they observed a dorsal enrichment of expression of *cnga3a* [58], like what we have observed in previous studies of the transducin subunit genes, *gnb3b* and *gngt2b* [11]. When we interrogated a developing zebrafish scRNA-seq atlas [59] for CNG genes, we could observe that all but *cnga2a*, *cnga2b* and *cnga4* were specific for photoreceptor cells, *cnga2a* did not show any expression and the others were expressed in olfactory neurons.

To conclude, expression data from non-bilaterian animals suggest that the formation of heterotetrameric channels in adults could be specific for cnidarians and bilaterians, possibly because of the presence of a nervous system of shared origin or the presence of a more complex nervous system in these lineages that require more fine-tuned modulation, something a more diverse pool of subunits would allow. Data from the tunicate *Ciona intestinalis* support a

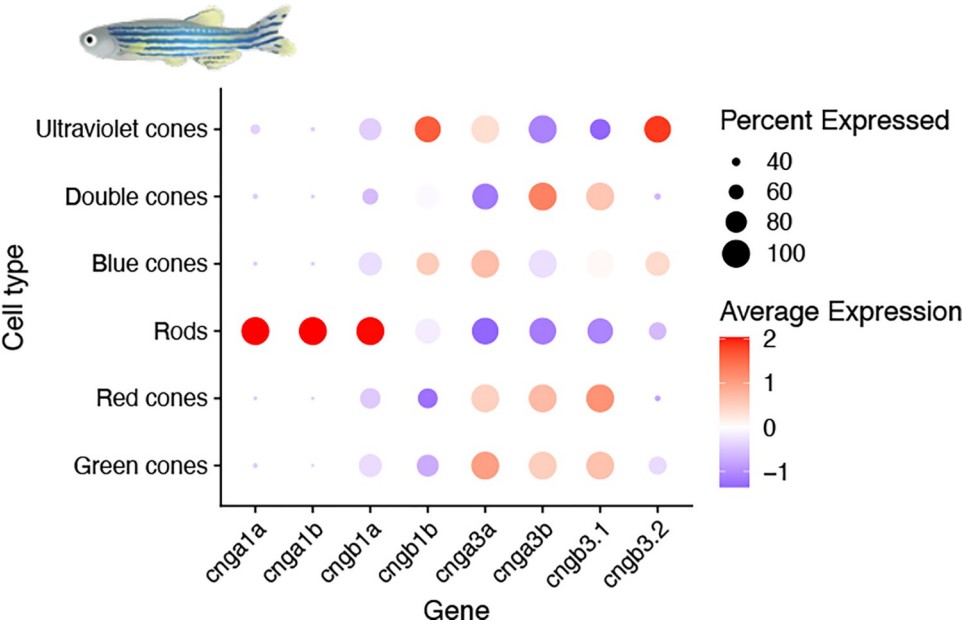

**Fig 10. CNG gene expression data from zebrafish.** Dot plot of the expression of zebrafish visual CNG genes in photoreceptor cell types in the retina. This data show that further specialization has occurred after the teleost-specific 3R event. Illustration of zebrafish was kindly provided by Dr. Daniel Ocampo Daza.

hypothesis of channel composition that is not the traditional vertebrate CNGA-CNGB type channel, but rather a CNGA-CNGD channel in our lineage, the chordates, in some neural cell types. Data from sponge and tunicates also suggest that CNGB might be able to form a functional homotetrameric channel at least outside of vertebrates. This together with the findings of several novel CNG genes invite more detailed studies to determine the function and subunit composition of channels related to these newly identified genes as well as their expression in various metazoan lineages. The analyses presented here reveals an ancient complexity of CNG channel genes that was reduced before the appearance of vertebrates. The two gene repertoire in the vertebrate ancestor was re-expanded to the six genes in the cyclostome and gnathostome ancestors, facilitating the differentiation of expression between cones, rods and other CNS functions. Furthermore, specializations in the teleost retina have been facilitated by retention of duplicates from mainly WGD events—adding further evidence for the important role of WGDs in eye evolution.

## Materials and methods

### Sequence identification and collection in chordates

Amino acid sequences of the CNGA and CNGB gene families from the species listed in S1 Table were downloaded from Ensembl (version 68) [60] and, when needed, updated to newer versions (see [61] for latest Ensembl paper). TBLASTN and BLASTP [62] searches either at Ensembl or NCBI were used to identify gene sequences that were missing in certain species. Additional sequences from other species were sometimes included to fill in gaps in the representation, for information about this see S1 Table. Identified genes that lacked annotation in the Ensembl genome browser were either annotated manually, by following sequence homology and splice donor/acceptor sites, or using gene prediction software like Augustus (3.2.2) [63,64] or the GENSCAN [65] web server available at: http://genes.mit.edu/GENSCAN.html.

Tables listing Ensembl gene IDs, Ensembl transcript IDs and the genomic locations of the CNGA and CNGB genes used are provided in S1 Table.

## Identification of CNG genes in actinopterygian fish

The human *CNGA4* amino acid sequence was used in a BLASTP search against the RefSeq protein database at NCBI with standard settings. All hit sequences (4063) were retrieved and subjected to a BLASTP search against the human proteome using BLAST+ 2.6.0 with -max_-target_seqs "1" and standard settings. Sequences that had a CNGA or CNGB gene as a best hit in the BLASTP search, were collected. Gene ID information for each protein sequence ID was downloaded from NCBI and each set of sequences was subjected to IsoSel [66] to extract the best transcript from each gene. The filtered output files were aligned using ClustalO [67] and the resulting alignments were inspected manually. The sequences that were too short (under one third of the average length of the sequences in the alignment), aligned poorly or appeared to belong to another gene family were removed from the analysis. Sequences labelled as "LOW QUALITY PROTEIN" (54 out of 870 of the CNGA sequences and 45 out of 491 of the CNGB sequences) were included in the analysis even though they might be pseudogenes, this since we wanted to disentangle the modes of duplication in this gene family and exclusion of these might have hidden duplications that have been lost to pseudogenization.

A time-calibrated phylogeny for all species included in the final actinopterygian fish CNG alignments was downloaded from timetree.org, only species that had this information available were included [68]. The final counts of sequences were mapped to this species phylogeny using iTOL [69].

## Identification of CNGA and CNGB genes in non-vertebrate eukaryotes

Human *CNGA4*, *CNGA3* and *CNGB1* amino acid sequences were used as queries in BLASTP searches against the NCBI nr database in metazoans, vertebrates excluded, with standard settings. All hit sequences were downloaded and used in a reciprocal BLASTP searches against the human proteome using BLAST+ 2.9.0 with -max_target_seqs "1" and standard settings. Sequences that had any CNG sequence as a best hit, i.e., putative CNGs, were then subjected to CD-HIT 4.8.1 [70,71] to remove duplicated sequences or alternative transcripts (designated as ≥ 95% sequence identity). The resulting consensus sequences were then subjected to a domain search using hmmscan 3.3.2 against the PfamA database.

Sequences with a similar domain architecture as the human CNGA proteins (with domains Ion_trans and cNMP_binding with e-values <0.01) were collected and added to the main family alignment of CNGA. An additional BLASTP search against the human proteome was performed to identify sequences that had something other than a CNG sequence among the top five hits, these sequences were excluded from the analysis. The sequences were trimmed as described in the alignment section. Sequences shorter than one third of the average length of the sequences in the main family alignment were removed. The resulting FASTA file was aligned as described in the multiple sequence alignment section and a phylogenetic tree was constructed as described in the phylogenetic analyses section.

Sequences with a cNMP_binding domain with an evalue <0.01 and that had CNGB as the highest hit in the reciprocal BLASTP search against the human proteome using BLAST+ 2.9.0 with -max_target_seqs "1" and standard settings, were added to the main family alignment of CNGB. The sequences were trimmed as described in the alignment section. Sequences shorter than one third of the average length of the sequences in the main family alignment were removed. The resulting FASTA file was aligned as described in the multiple sequence alignment section and a phylogenetic tree was constructed as described in the phylogenetic analyses section.

Sequences labelled as "LOW QUALITY PROTEIN" (19 out of 911 of the CNGA sequences and 0 out of 379 of the CNGB sequences) were included in the analysis even though they might be pseudogenes, this since we wanted to disentangle the modes of duplication in this gene family and exclusion of these might have duplications that have been lost to pseudogenization.

In addition, a BLASTP search was performed against the nr database using the same sequences, against all eukaryotes, with metazoans excluded. Sequences that had any CNG sequence as best was then subjected to CD-HIT 4.8.1 [70,71] to remove duplicate sequences or alternative transcripts (designated as ≥ 95% sequence identity). The resulting consensus sequences were then subjected to a domain search using hmmscan 3.3.2 against the PfamA database. Hmmscan results were then investigated for sequences with the CLZ domain.

Finally, NCBI protein IDs were used to retrieve NCBI taxonomy identifiers to identify which eukaryote groups that have either putative CNGA gene or putative CNGB genes or both.

## Multiple sequence alignments

Amino acid sequences collected from the *CNGA* and *CNGB* families, as well as from the identified neighboring gene families, were placed in gene family specific FASTA files and aligned using ClustalO [67] with standard settings. Alignments were inspected and sequences edited manually where necessary. Incomplete sequences of vertebrate sequences were either extended manually if possible or removed from the alignment if they were shorter than one third of the average length of the sequences in the alignment. The CNGA sequences in the main family alignment were trimmed in the beginning right up to the human CNGA1 "VVID" motif and after the "EYPD" motif in the same sequence. The CNGB sequences in the main family alignment were trimmed until the "NLMY" motif in human CNGB1 and after the "GTPK" motif in the same sequence. The trimming was done to remove the variable N and C terminal regions that are often missing and/or difficult to manually extend.

## Phylogenetic analyses

Alignments for all families except for the metazoan CNGA sequences were used for substitution model prediction and phylogenetic tree reconstructions using a local installation of IQ-TREE using aLRT and ultra-fast bootstrapping [72–74]. The following settings were used: -t BIONJ -quiet -keep-ident -bb 10000 -alrt 10000 -m TEST. For the metazoan CNGA sequences, the Galaxy [75] server (https://usegalaxy.eu) was used to run IQ-TREE using standard settings and -m TEST as well as -bb 10000 and -alrt 10000.

The resulting ML trees were rooted with nw_reroot from Newick utilities [76] and displayed using ggtree R package [77] or rooted and displayed using iTOL [69] (available at https://itol.embl.de). Nodes considered well supported were labelled with a filled red circle with the criteria ≥80% aLRT and ≥95% Ufbootstrap support (based on the IQ-TREE manual), for all trees except for the metazoan CNGA tree. The trees of the actinopterygian fish CNG sequences were either rooted with the CNGA4 sequences (CNGA tree) or rooted by one of the subtype clades (CNGB tree). The metazoan CNGA the tree shown, is the consensus tree and the node support with ≥90% are labelled with a filled red circle.

## Comparative synteny analysis of the bony vertebrate *CNGA* and *CNGB* regions

Lists of genes located in regions 5 mega base pairs (Mb) upstream and downstream of the spotted gar (LepOcu1 assembly) CNGA genes were retrieved from Ensembl (version 74). *CNGA3*

and *CNGA4* are located so close to each other on LG17 in spotted gar that the 5 Mb regions overlap. In this case therefore, a region 5 Mb upstream of one and 5 Mb downstream of the other gene were used in the selections of neighboring gene families. Genes belonging to Ensembl protein families with members on all three of the CNGA carrying spotted gar linkage groups were selected for further phylogenetic analysis (see S1-S7 Figs in S1 File for the phylogenetic trees). For the CNGB genes lists of genes located in regions of 10 Mb upstream and downstream of the spotted gar CNGB genes were retrieved. Genes belonging to Ensembl protein families with members on both CNGB carrying spotted gar linkage groups were selected for further phylogenetic analysis (see S8-S17 Figs in S1 File for phylogenetic trees).

## Comparative synteny analysis of the elasmobranch *CNGA* and *CNGB* regions

Amino acid sequences from annotations of the small-spotted catshark were compared to the amino acid sequences of spotted gar using Proteinortho [78] at the Galaxy [75] server (https://usegalaxy.eu). Tables containing the spotted gar gene and its small-spotted catshark ortholog were made and they were colored based on spotted gar chromosome. If a spotted gar gene seemed to be missing, we performed best reciprocal BLASTP searches to identify the missing genes. The data for the small-spotted catshark can be accessed through NCBI BioProject: PRJEB35945.

## CNG gene expression

The DotPlot showing the expression of phototransduction specific CNG genes in zebrafish photoreceptors is based on GEO data from Ogawa and Corbo (2020) (gene expression omnibus accession number GSE175929). The figure shows the same data shown in their Fig 3C but plotted differently for clarity using Seurat v4 [79]. The information about the expression of phototransduction specific CNG genes from zebrafish cells during development were retrieved from the following website: http://zebrafish-dev.cells.ucsc.edu based on analyses from [59]. Expression data from *Ciona intestinalis* larval central nervous system was retrieved from https://singlecell.broadinstitute.org/single_cell/study/SCP454/comprehensive-single-cell-transcriptome-lineages-of-a-proto-vertebrate [80]. Data from non-bilaterian metazoans were retrieved from https://tanaylab.github.io/old_resources/ [40,41].

## Supporting information

**S1 File. File containing all supplementary figs and their descriptions.**
(PDF)

**S1 Table. Genome assemblies, sequence identifiers and chromosomal locations of the genes included in the phylogenetic analyses of the main and neighboring gene families.** NCBI protein accession numbers and species names of actinopterygian fish sequences included in the analyses.
(XLSX)

**S2 Table. Species, NCBI protein accession number, chromosome, and possible mode of duplication of genes located on the same chromosome within one actinopterygian fish species as well as locations of lamprey CNG hits in species with genome assemblies available on NCBI.**
(XLSX)

## Acknowledgments

We would like to thank Christina Bergqvist and Jenny Widmark for help with the initial phylogenetic analyses and valuable discussions. We would also like to thank Daniel Ocampo Daza for the animal drawings used in Figs 6B, 6C and 7E.

## Author Contributions

**Conceptualization:** David Lagman, Xesús M. Abalo, Dan Larhammar.

**Data curation:** David Lagman, Helen J. Haines.

**Formal analysis:** David Lagman, Helen J. Haines.

**Funding acquisition:** Xesús M. Abalo, Dan Larhammar.

**Investigation:** David Lagman, Helen J. Haines, Xesús M. Abalo, Dan Larhammar.

**Methodology:** David Lagman.

**Project administration:** Xesús M. Abalo, Dan Larhammar.

**Supervision:** Xesús M. Abalo, Dan Larhammar.

**Visualization:** David Lagman.

**Writing – original draft:** David Lagman.

**Writing – review & editing:** David Lagman, Helen J. Haines, Xesús M. Abalo, Dan
    Larhammar.

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
