## [Decision Letter · Decision Letter 0]

24 Oct 2022

PONE-D-22-27051Ancient multiplicity in cyclic nucleotide-gated (CNG) cation channel repertoire was reduced in the ancestor of Olfactores before re-expansion by whole genome duplications in vertebratesPLOS ONE

Dear Dr. Lagman,

Thank you for submitting your manuscript to PLOS ONE. After careful consideration, we feel that it has merit but does not fully meet PLOS ONE’s publication criteria as it currently stands. Therefore, we invite you to submit a revised version of the manuscript that addresses the points raised during the review process.

Please include most if not all the different suggestions of reviewers since I find them quite useful. If you decide to not include some of the suggestions, please specify why. Please submit your revised manuscript by Dec 08 2022 11:59PM. If you will need more time than this to complete your revisions, please reply to this message or contact the journal office at plosone@plos.org. Please include the following items when submitting your revised manuscript:A rebuttal letter that responds to each point raised by the academic editor and reviewer(s). You should upload this letter as a separate file labeled 'Response to Reviewers'.A marked-up copy of your manuscript that highlights changes made to the original version. You should upload this as a separate file labeled 'Revised Manuscript with Track Changes'.An unmarked version of your revised paper without tracked changes. You should upload this as a separate file labeled 'Manuscript'.

We look forward to receiving your revised manuscript.

Kind regards,

Hector Escriva, PhD

Academic Editor

PLOS ONE

Journal Requirements:

Reviewers' comments:

Reviewer's Responses to Questions

**Comments to the Author**

1. Is the manuscript technically sound, and do the data support the conclusions?

Reviewer #1: Yes

Reviewer #2: Yes

2. Has the statistical analysis been performed appropriately and rigorously? 

Reviewer #1: Yes

Reviewer #2: Yes

3. Have the authors made all data underlying the findings in their manuscript fully available?

Reviewer #1: No

Reviewer #2: Yes

4. Is the manuscript presented in an intelligible fashion and written in standard English?

Reviewer #1: Yes

Reviewer #2: Yes

5. Review Comments to the Author

Reviewer #1: Lagman et al present a very detailed and thorough study of the evolution of Cyclic nucleotide-gated cation channels. The authors do a very nice job of putting the results in context both of Metazoan genome evolution, and of the evolution of vision. Overall this is a good focused study of molecular evolution, and I only have a few comments.

Detailed comments:

The authors use the term "invertebrate" many times. This is misleading because as invertebrates are paraphyletic, there is no generalization which covers their evolution and does not also concern vertebrates. When the aim is to study more general metazoan evolution, relaxing the focus on vertebrates, the authors should specify this, and when specific groups are concerned (e.g. metazoans which do not have eyes), they should also specify it. In each case of use of the tern invertebrates, a better term can be found by asking which evolutionary question is being answered, and formulating it clearly.

In the Introduction, the opening statements need references, especially the first sentence. I don't think that Darwin mentionning the eye in 1858 would be sufficient. Is there evidence that vision was one of the most intensively studied topics?

p. 5, the last paragraph on insight into medical conditions is neither convincing nor needed to make the point of the manuscript, I suggest removing it.

p. 11 and following: it would be helpful to present in a figure or table which of the species or lineages sampled use vision "as an important sense" (e.g. most vertebrates, most insects, cephalopods), which use it as a secondary sense, and which do not have vision.

It would be interesting to compare the vertebrate and especially fish synteny results with the results computed in the specialised tool Genomicus: https://www.genomicus.bio.ens.psl.eu/genomicus/

In the methods, please specify the flavor of Blast (sometimes specified as BlastP, sometimes just "Blast"), the version and the parameters used.

Why use reciprocal Blast rather than an ortholog cluster algorithm on all potential homologous genes from the first round of Blast hits?

Figure 5: it is not possible to distinguish the closely related colors, which makes the figure very difficult to read and interpret.

Legends of Fig 6-8: please specify the color code the first time it is used, i.e. Fig 6.

Typos:

line 74 "an origin"

line 539 "extend" not "extending"

line 451 should be same sentence grammatically.

The following data is "not publicly available" on Figshare: CNGB_actinopterygian_tree.

Reviewer #2: The manuscript presents a detailed and very interesting study of the evolution of CNGA and CNGB genes, and clarify duplication and loss processes that led to the current gene repertoires in metazoan species. This study is very complete, combining the analysis of a huge amount of publicly available genomic and transcriptomic data. The literature used as reference is rich and its combination with a well written introduction helps a lot in understanding the results and the discussion. However, in my opinion, the manuscript would gain in clarity after some modifications of the main text and figures and a restructuring.

Suggestions concerning figures:

- I would suggest the authors to color gene phylogenies branches according to the species clade they belong to. This would allow a better understanding of the results, especially for Figure 2, Figure 6 and Figure 7. This is not needed for figure 7 as species clades are already indicated.

- Merge Figure 1 and Figure 3 to a unique figure. One idea: Keep the figure 3 as it is, but adding a phylogenetic tree connecting species clade at the left of the figure (in the same way as what was done for figure 1), and add a column for the presence/absence of CNGB gene. To keep the information about the presence/absence of the cNMP binding domain, and the CLZ domain, a symbol in the boxes can be added.

- Figure 5: Add species names in addition to the silhouettes.

- Figure 5-E: Removing the circles when no expression is detected, instead of small circles, would improve the figure readability.

- Figure 8: Indicate what the number written in the gene boxes represent, either in the figure or in the legend.

- Figure 9: Indicate with an arrow the teleost specific whole genome duplication, in the branch leading to the zebrafish.

- Move Table 1 to supplementary table

Suggestions concerning the main text:

- Line 46: “Led to us to” -> “Led us to”

- Line 137 to line 146:

Replace “We identified five clades” to “We identified five CNGA type gene clades”

Here is a proposition for the rest of the paragraph, that would gain in clarity by being rewritten:

“We propose a naming for these five clades : - CNGA (named upon traditional CNGA sequences) in which most genes have a CLZ domain and containing representative sequences of vertebrates and sponge ; - CNGE which is only found in cnidarian ; - CNGF which is present in bilaterian and cnidarian but which have been lost in deuterostomes ; -CNGC present in both Ctenophore, Cnidaria and some Bilateria clades ; -CNGD which is found in most of metazoan excepted in sponges and placozoans “

- Line 184 to line 186: I believe you in what you have written, but the decoupling expression between CNGA and CNGB seems unclear to me in the Figure 5-B, where CNGA expression seems very low overall compared to the CNGB expression. This could be clarified by modifying the X-axis of the CNGA expression plot. I don’t see any problem of having different X-axis values for CNGA and CNGB.

- Line 205: Why is the green spotted pufferfish not on the figure 8?

I would also suggest to remove this information from the main text and to place it in the figure 8 legend as it is not discussed and not useful for the study conclusions. Same goes for the Japanese pufferfish information on Line 253 to 256.

- Line 218 to 222: Redundancy of results with lines 239 to 256.

- Line 226: Change “To investigate the CNG gene repertoire further in a group of vertebrates where vision is a very important sense” to “To further investigate the evolution of CNG genes repertoire in a group of vertebrates with a rich and dynamic opsin gene repertoire (ref1, ref2, ref3) …”

Ref 1 : https://www.ncbi.nlm.nih.gov/pmc/articles/PMC4617963/

Ref 2 : https://www.science.org/doi/10.1126/science.aav4632

Ref 3 : https://doi.org/10.1146/annurev-cellbio-120219-024915

I disagree with the postulate that vision is a very important sense in actinopterygian. First, I believe that the rich opsin gene repertoire of these species is not due to a better vision than other vertebrates, but is due to the highly different light environments they face. Also, many fish species don’t rely at all on vision (cavefishes for example) and many species seems to rely much more on olfaction and taste for feeding, orientation or conspecific recognition. Thus, I also suggest to change this statement made on Line 118

- Line 230: Replace “sturgeons that have seven genes” by “sturgeons and paddlefishes that have seven genes”

- Line 233: I would remove all those details about local duplications and simply write “in general, teleost fish have seven CNGA genes, with exceptions e.g. cyprinids and salmonids have experienced extra lineage specific WGDs, and some clade specific local duplications”

- Line 332: “rhe” -> “the”

- Line 381: If I understand correctly, you believe that the duplication of CNGB3 in teleost did not arise from the WGD, but by a local duplication. This seems unclear to me. What arguments do you have to differentiate between such a local duplication and the effect of WGD followed by a rearrangement?

- Line 389-390: The fact that all those species only have one CNGA4 excepted for Carassius auratus is interesting. Could you look at if in this species, one of the CNGA4 copy is a pseudogene but have not yet been erased from the genome? Also, I believe that it would be worth to mention somewhere in the manuscript or in the methods, that you don’t consider differences between complete genes and pseudogenes in your gene copy numbers. This is clearly visible on the datasupp2, where some genes are annotated as “LOW QUALITY PROTEIN” which are usually genes with frameshifts or premature stop codons in NCBI. This does not change your results and discussion as you focus on the overall gene repertoire and its evolution at a high evolutionary timescale (And it would need a lot of additional work to make a proper annotation of pseudogenes).

I also suggest to restructure the manuscript with this chapter order: 2.1 ; 2.3 ; 2.4 ; 2.5 ; 2.6 ; 2.7 ; 2.2. This would allow to first explain the evolution of CNGA and CNGB genes at the genomic level, and to then investigate their expression. With the actual chapter order, the transcriptomic analysis kind of break the flow by being placed in between the genomic investigations.

- Line 479: Please add details on the methods used for your manual annotation of genes.

Methods sections 5.6 and 5.7: Did you make the synteny figures with a software? If so, indicate which software was used.

Congratulations to the authors for this study, hoping that my suggestions, and those of the other reviewers, will help for the publication of this manuscript.

6. PLOS authors have the option to publish the peer review history of their article (what does this mean?). If published, this will include your full peer review and any attached files.

Reviewer #1: **Yes: **Marc Robinson-Rechavi

Reviewer #2: **Yes: **Maxime Policarpo

---

## [Author Response · Author response to Decision Letter 0]

22 Nov 2022

Below is our answers to the comments from the editor:

This has been adressed in the revised manuscript.

The grant information has been updated to contain grant numbers, however I can't seem to update the Financial disclosure section to contain the grant numbers. below is the updated Financial disclosure statement:

This work was supported by two grants to DLar from Vetenskapsrådet (C0452101) and Carl Tryggers Stiftelse för Vetenskaplig Forskning (CTS 09:210) (https://www.vr.se and https://www.carltryggersstiftelse.se respectively) and one grant to XMA from Stiftelsen Olle Engkvist Byggmästare (468163019) (https://engkviststiftelserna.se). The funders had no role in study design, data collection and analysis, decision to publish, or preparation of the manuscript.

This has been adressed in the revised manuscript.

---

## [Decision Letter · Decision Letter 1]

12 Dec 2022

Ancient multiplicity in cyclic nucleotide-gated (CNG) cation channel repertoire was reduced in the ancestor of Olfactores before re-expansion by whole genome duplications in vertebrates

PONE-D-22-27051R1

Dear Dr. Lagman,

We’re pleased to inform you that your manuscript has been judged scientifically suitable for publication and will be formally accepted for publication once it meets all outstanding technical requirements.

Kind regards,

Hector Escriva, PhD

Academic Editor

PLOS ONE

Additional Editor Comments (optional):

Reviewers' comments:

Reviewer's Responses to Questions

**Comments to the Author**

1. If the authors have adequately addressed your comments raised in a previous round of review and you feel that this manuscript is now acceptable for publication, you may indicate that here to bypass the “Comments to the Author” section, enter your conflict of interest statement in the “Confidential to Editor” section, and submit your "Accept" recommendation.

Reviewer #1: All comments have been addressed

Reviewer #2: All comments have been addressed

2. Is the manuscript technically sound, and do the data support the conclusions?

Reviewer #1: Yes

Reviewer #2: Yes

3. Has the statistical analysis been performed appropriately and rigorously? 

Reviewer #1: Yes

Reviewer #2: Yes

4. Have the authors made all data underlying the findings in their manuscript fully available?

Reviewer #1: Yes

Reviewer #2: Yes

5. Is the manuscript presented in an intelligible fashion and written in standard English?

Reviewer #1: Yes

Reviewer #2: Yes

6. Review Comments to the Author

Reviewer #1: I thank the authors for the excellent work doing this revision.

One small comment: line 176, "in bilaterians and cnidarians" could be "in protostomes".

Reviewer #2: The major concerns that I had in the initial submission have been correctly addressed in this new manuscript, and I thank the authors for their answers to my various questions and for restructuring the manuscript and some figures. The manuscript now look great and I recommend it to be published in PLOS ONE.

7. PLOS authors have the option to publish the peer review history of their article (what does this mean?). If published, this will include your full peer review and any attached files.

Reviewer #1: **Yes: **Marc Robinson-Rechavi

Reviewer #2: **Yes: **Maxime Policarpo

---

## [Editor Report · Acceptance letter]

20 Dec 2022

PONE-D-22-27051R1 

­­Ancient multiplicity in cyclic nucleotide-gated (CNG) cation channel repertoire was reduced in the ancestor of Olfactores before re-expansion by whole genome duplications in vertebrates 

Dear Dr. Lagman:

I'm pleased to inform you that your manuscript has been deemed suitable for publication in PLOS ONE. Congratulations! Your manuscript is now with our production department. 

Kind regards, 

on behalf of

Dr. Hector Escriva 

Academic Editor

PLOS ONE